# Needs of Families with Children with Cerebral Palsy in Latvia and Factors Affecting These Needs

**DOI:** 10.3390/jpm10030139

**Published:** 2020-09-22

**Authors:** Dace Bertule, Anita Vetra

**Affiliations:** Department of Rehabilitation, Riga Stradins University, LV-1007 Riga, Latvia; anita.vetra@rsu.lv

**Keywords:** cerebral palsy, children, family needs

## Abstract

In order to provide targeted support to families who are raising children with developmental disorders, it is important to study the family needs and to understand circumstances that may affect them. The aim of this study was to identify the needs of the families with preschool children with cerebral palsy, and study how these needs relate to factors associated with families, children and rehabilitation services. Descriptive analysis showed that families living in Latvia most often need information, social and financial support and coordination of services, and they also need financial support to cover the costs of child care and treatment. The results of the data analysis support the hypothesis that factors characterising families, children with cerebral palsy and rehabilitation services affect the needs of the families with preschool children with cerebral palsy living in Latvia, and the unique impact of these factors depends on the type of needs. Regression analysis revealed that the most important factors affecting the needs of families were related with the socio-economic situation, as well as the support of peers and professionals. The availability and regularity of rehabilitation services, limitations to the child’s functions and health impairments were factors that affected family needs to a lesser extent.

## 1. Introduction

Cerebral palsy (CP) describes a group of permanent disorders in the development of movement and posture, causing activity limitation, which are attributed to non-progressive disturbances that occurred in the developing fetal or infant brain. The motor disorders of CP are often accompanied by disturbances of sensation, perception, cognition, communication and behaviour, by epilepsy and by secondary musculoskeletal problems [1].

Cerebral palsy is the most common cause of physical disability among children, having a potentially negative impact on the quality of life and involvement in the community not only for the child, but also for the whole family [2]. Meanwhile, the psychic, emotional and financial welfare of the child’s closest relatives is a significant factor effecting the child’s development [3,4]. Therefore, health care providers should be aware that, in order to promote successful child development, such kinds of service are needed that simultaneously target the improvement of the functions of the child and their family.

Services based on the principles of family-centred care have been recognised as the most efficient services in the work with families with children with developmental disorders [5]. There have been findings that show that, in this way, the best results can be achieved in improving the child’s functioning and meeting the family’s specific needs, as well as increasing the quality of life [5,6,7,8].

A significant aspect of family-centred health care is the study of family needs and priorities [9]. Meanwhile, information about factors influencing family needs would help us to understand the potential needs of each individual family, and thus offer the support that is most important for the particular family. Based on the ecological theory of family systems and human development, the factors affecting potential family needs are determined by the features and characteristics of every individual in the family, the family as a whole, and the environment [10]. Therefore, information about the characteristics of the child, the family and services that can possibly influence the family’s needs would promote the provision of family-centred services to families raising a child with cerebral palsy, thus providing the best possible support for the child’s development and allowing the improvement of quality of life for the family.

The aim of the study was to identify the needs of the families with preschool children with cerebral palsy, and to study how these needs relate to the factors characterising families, children and rehabilitation services. The objectives of the study were as follows:To identify the specific needs of the families with preschool children with cerebral palsy;To assess and analyse the correlation between the needs of the families with children with cerebral palsy and the factors characterising the demographic and socio-economic situation of the families, as well as the factors characterising children with cerebral palsy, and the factors characterizing the availability of rehabilitation services and cooperation between the family and rehabilitation service providers;To assess the impact of the factors characterising families, children and rehabilitation services on the needs of the families with children with cerebral palsy.

### Hypothesis of the Study

The needs of families raising preschool children with cerebral palsy living in Latvia are affected by the factors characterising families, children with cerebral palsy and rehabilitation service providers, and the unique impact of these factors depends on the type of family needs.

## 2. Materials and Methods

A conceptual model of the determinants of family needs (Figure 1) was developed based on the following:the ecological theory of family systems and human development, determining that family needs can be influenced by peculiarities of the individual, the whole family and the surrounding environment [10];the conceptual model of the determinants of needs of families raising children and youth with cerebral palsy proposed by Almasri and colleagues [7];comprehensive study of the literature, collecting information about the types of the needs of families raising children with cerebral palsy, and identifying the possible factors affecting these family needs.

### 2.1. Participants and Procedure

The study covered 234 families living in Latvia who, during the period of collecting data, had a preschool-aged child (in Latvia, the preschool age is from 2 to 7 years) diagnosed with cerebral palsy. Contact information of the families was obtained through the patients’ register, and the medical documentation of the Children’s Clinical University Hospital, National Rehabilitation Centre “Vaivari” and “Mēs esam līdzās” organisations. If during the selection process the families visited any of the above-mentioned institutions, they were addressed. If the family met the criteria and agreed to participate in the study, a functional assessment of the child was conducted, and the child’s primary caretaker filled in the questionnaires included in the assessment methods. If the family did not visit any of the above-mentioned institutions, the family was addressed by telephone and informed about the study, and a meeting was arranged if the family agreed to participate. In most of the cases, a period was selected when the child had an ongoing rehabilitation course in any of the above-mentioned institutions. Seven families were visited at the place of their residence.

Criteria for the study:the child’s main clinical diagnosis—cerebral palsy (according to ICD-10 classification: G80);the child’s age is 2–7 years;the family’s permanent place of residence is in the Republic of Latvia;the child’s primary caretaker has agreed to participate in the study.

Families were not included in the study if any of the exclusion criteria could be applied, as follows:the child with cerebral palsy had additional disorders that might seriously affect the quality of life of the child and the family (e.g., severe autism spectrum disorder, cystic fibrosis, malignant tumour);any of the family members has severe health disorders that might affect the quality of life of the child and the family (e.g., progressing neuromuscular disorder, severe autism spectrum disorder, cystic fibrosis, malignant tumour, severe mental retardation);in the past two years the family had not used the services of the state or local government (e.g., medical or social rehabilitation, preschool educational institutions, day care centre) for the child with cerebral palsy.

Assessment of the child’s functions and questionnaires took 90 min on average. The author of the study was present to answer any questions that could arise during the filling in of the questionnaire.

As standardised self-evaluation questionnaires were used in the study as an assessment method, and the originals of the questionnaires are in English and had not been translated into Latvian before the study, the questionnaires were translated in line with the recommendations of the World Health Organization (WHO, Process of translation and adaptation of instruments), and a pilot study was conducted to test the reliability of the Latvian translation of the questionnaires.

### 2.2. Measures

#### 2.2.1. Outcome Measure

In order to identify family needs, the child’s primary caretaker filled in the Family Needs Survey [11].

Originally, the survey consisted of 35 items. The items were grouped in 6 subscales according to the type of needs: (1) Needs for Information; (2) Needs for Support; (3) Explaining to Others; (4) Community Services; (5) Financial Needs; (6) Family Functioning. In terms of the needs that might be more specific in our country, 1 item was removed, and 6 items were added with the written permission of the authors, for a total of 41 items. The item that was removed was “Getting child care in church/synagogue”. The items that were included are as follows: “Finding information about planning child’s wellbeing in future”, “Finding information about future education”, “Explaining my child’s condition to professionals”, “Locating rehabilitation services”, “Coordinating medical, developmental, educational, and other community services”, and “Paying for home modification”.

The internal consistency (Cronbach alpha) and test–retest reliability (ICC 2.1) of the translated and modified version of FNS were tested in advance of the original study. The Cronbach alpha coefficient for all items was 0.82 and ranged between 0.71 and 0.89 for each subscale. ICC 2.1 for all items was 0.89 and the coefficient varied from 0.72 to 0.98.

The items were rated on a 3-point scale. The response options were as follows: 1 = I definitely do not need help with this, 2 = Not sure, 3 = I definitely need help with this. For the purpose of this study, only the items that were scored with number 3 (I definitely need help with this) were considered to be unmet family needs, and were scored in order to get the total number of family needs and the number of each type of need for data analysis.

As two of the areas of needs (‘need for support’ and ‘need for family functioning’) are related to the wish for support, then, like Almasri’s study [8], they were considered as a single bloc of needs—the need for support.

#### 2.2.2. Determinants Measures

##### Questionnaire for General Information

The authors of the study developed a questionnaire to obtain information about the age and education of the child’s primary caretaker, the family structure, socio-economic situation, the place of residence, the child’s overall health condition, as well as the availability of rehabilitation services in the place of residence and the regularity of the service provision.

##### Measure of Processes of Care: MPOC-20

The Measure of Processes of Care (MPCO-20) is a 20-item self-report about parental perception about the extent to which the health services that they and their child receive are family-centred. The survey includes five subscales:Enabling and Partnership;Providing General Information;Providing Specific Information;Co-ordinated and Comprehensive Care;Respectful and Supportive Care.

The items are rated on the Likert scale, ranging from 1 to 7 (1 = the described event or situation was not perceived; 7 = the described event or situation was perceived to a very great extent) [12].

This study made use of a version of the MPOC-20 that was translated into Latvian. The internal consistency (Cronbach’s alpha) and test–retest reliability (ICC 2,1) of the translated version of the MOPC-20 were tested before the original study. The Cronbach’s alpha co-efficient for all 20 items was 0.87 and varied between 0.77 and 0.94 for each subscale. The ICC 2,1 for all items was 0.94, and the co-efficient varied between 0.93 and 0.96.

##### Perceived Stress Scale-10

The Perceived Stress Scale (PSS) measures to what extend an individual has felt stressful situations during the past month. The scale includes 10 positive and negative statements that have to be answered on a Likert scale. The possible answers are: 5—very often, 4—quite often, 3—sometimes, 2—almost never, and 1—never. In order to obtain the overall stress perception indicator, the positive statements received a reversed score (i.e., if a question is scored “5”, the score turns into “1”, etc.), then all scores were added up, and the total perceived stress indicator was obtained. The higher the indicator, the higher the perceived stress level [13].

The study used the Latvian translation of the scale developed by Ieva Stokenberga for her Doctoral Thesis “The Role of Humour in the Process of Overcoming Stress” with the permission of the author and the Psychology Department of the University of Latvia. The internal consistency of the Latvian translation of the Perceived Stress Scale was Cronbach alpha = 0.83.

##### The Gross Motor Function Classification Scale (GMFCS)

GMFCS is a five-level classification system used to describe gross motor function for children with cerebral palsy. The assessment and classification of motor functions were conducted in line with the age groups. A description of the assessment of motor functions for five age groups has been developed, as follows: up to 2 years of age, 2–4 years, 4–6 years, 6–12 years, and 12–18 years. The assessment is conducted while observing the child’s abilities while sitting, changing positions and moving, and the performance determines which GMFCS level corresponds to the child’s motor functions [14]. A short description of GMFCS levels is included in Table 1.

##### The Communication Function Classification System (CFCS)

CFCS, developed by Hidecker and colleagues, is a five-level classification system characterising communication performance that initially was developed for use with children with cerebral palsy [15], but recently it has been approved for use also in other cases with children with communication disorders [16]. In order to determine which CFCS level the child’s communication performance corresponds to, the child’s communication with the relatives as well as unfamiliar people was observed. The child’s ability to receive and send information irrespective of the way it is being done was assessed. The child may use language, eye contact, gestures, communication devices, etc., for communication. CFCS levels are described in Table 2.

The data used in the study analysis and their description are presented in Table 3.

### 2.3. Statistical Analysis

The descriptive statistical methods were used for data analysis (average values, standard deviation, minimum and maximum values), and the frequency of occurrence of particular data was analysed. The distribution of data was tested using histograms.

In order to study the reliability and distinctiveness of the impact of different factors on one model (family needs) which consists of several latent variables, multiple linear regression analysis was conducted, performing the following steps:Dependent variables and independent quantitative variables were tested for normal distribution. If the dependent variable did not meet the normal distribution requirements, it was logarithmically scaled;In order to discover whether there is correlation between the dependent and independent variables, depending on the type of the analysed data and compliance with normal distribution, Pearson or Spearman correlation analysis was conducted. Only those independent variables that statistically reliably (*p* < 0.05) correlated with the dependent variable were included in the regression analysis;If the independent variable was nominal with several categories, it was recoded in dummy variables;Multiple linear regression analysis was conducted, including the selected independent variables. Enter method was used for regression analysis, but Forward and Backward methods were used to compare the results;The best model was selected, comparing the models with F test. If two models differed considerably (*p* < 0.05), the model with a higher R^2^ was selected. If the models did not differ substantially, the model with a lower number of explanatory variables was selected;Every end-model that explained the result best was tested for collinearity, linearity and normal distribution requirements;Factors that were included in the final model and were statistically significant were one by one excluded from the model, and the obtained R^2^ changes were used in describing the factors’ unique contributions to the model.

## 3. Results

### 3.1. Participants of the Research

As many as 259 children, aged 2–7, with G80 diagnosis were identified in the archives and databases of the National Rehabilitation Centre “Vaivari”, the Children’s Clinical University Hospital, and the organisation “Mēs esam līdzās”. The author was not able to establish contacts with five families and the representatives of 20 families refused to participate in the study, and therefore 234 families were engaged in the study (a child with cerebral palsy and the child’s primary caretaker). The detailed characteristics of the respondents and families are described in Table 4.

The Perceived Stress Scale (PSS) of respondents varied from 6 points to 42 points, with 24.4 points on average (SD = 7.1).

THE Assessment of support provided to respondents by family members (FSS family) was in the range between 5 points and 22 points, with 12.5 points on average (SD = 3.5), while the assessment of support from friends and social groups (FSS informal) varied from 8 points to 26 points, with 18.8 points on average (SD = 3.5). The assessment of support provided by professionals (FSS professionals) ranged from 4 points to 15 points, with 8.8 points on average (SD = 2.5).

The average age of children with cerebral palsy was 4.8 years (SD = 1.7). A detailed description of children with cerebral palsy is given in Table 5.

### 3.2. Description of Availability and Regularity of Rehabilitation Services

As many as 149 respondents, or 63.7% (95% CI: 58.7–68.1), said that they have rehabilitation services available at the place of residence, while 85 respondents, or 36.3% (95% CI: 32.1–39.8), said that there is no appropriate rehabilitation service available close to their place of residence.

Analysing the regularity of rehabilitation, the authors discovered that 60 children, or 25.6% (95% CI: 20–31.2), receive government-funded rehabilitation services on a regular basis, 93 children, or 39.7% (95% CI: 33.4–46.1), undergo rehabilitation in courses several times a year, and 81 children, or 34.6% (95% CI: 28.4–40.7), undergo a rehabilitation course once a year or less frequently.

### 3.3. The Results of Measure of Processes of Care (MOPC-20)

The analysis of results of the Measure of Processes of Care (MOPC-20) revealed that rehabilitation service providers partly follow the principles of family-centred services. The respondents were most positive about such principles as “Respectful and Supportive Care” (mean = 4.84; SD = 1.8), “Enabling and Partnership” (mean = 4.65; SD = 1.29) and “Co-ordinated and Comprehensive Care” (mean = 4.62; SD = 1.17). Respondents were more negative about principles related to the provision of information—“Providing Specific Information” (mean = 3.62; SD = 1.21) and “Providing General Information” (mean = 3.32; SD = 1.20).

### 3.4. The Results of Family Needs Survey (FNS)

Analysing the overall results of the Family Needs Survey, the authors discovered that every respondent marked at least three statements regarding the needed assistance. Detailed information about the Family Needs Survey is available on Table 6.

The average number of needs marked by respondents was 17.1 (SD = 7.1), varying from 3 to 38. The score values, characterising respondent needs in separate types of needs, are depicted in Table 7.

For further analysis, two types of needs (“Needs for Support” and “Family Functioning”) were viewed together, forming “Needs for Support”, wherein the average number of needs was 4.3 (SD = 2.1), varying from 0 to 10. As the number of needs in the subscale “Explaining to Others” was low, the influencing factors were not viewed for this type of needs.

### 3.5. Results of Correlation Analysis

#### 3.5.1. Correlation between Family Needs and Child’s Characteristics

The age of the child was not related to family needs. Other factors characterising children with cerebral palsy were related to at least one type of family needs. Correlations that were statistically significant are depicted in Table 8.

#### 3.5.2. Correlations between Family Needs and Family Characteristics

No statistically significant correlation was found between family needs and the age and family status of the child’s primary caretaker. Other factors characterising the family were related to at least one type of family needs. Correlations that were statistically significant are depicted in Table 9.

#### 3.5.3. Correlations between Family Needs and Service Characteristics

A correlations analysis revealed that all the factors characterising the rehabilitation services are related to at least one type of family needs. Correlations that were statistically significant are depicted in Table 10.

### 3.6. Results of Multiple Linear Regression Analysis

In order to explain the factors influencing family needs, multiple linear regression analysis was used. Five family needs models were distinguished and viewed for the analysis, as follows:the model that explains overall family needs;the model that explains family needs for support;the model that explains family needs for community services;the model that explains the financial needs of the family;the model that explains family needs for information.

All final models met the requirements of collinearity, linearity and normal distribution of regression analysis.

#### 3.6.1. Model that Explains Overall Family Needs

The total number of needs (FNS) marked by families was used as the dependent variable. In the initial model, five factors characterising children with cerebral palsy (GMFCS level, CFCS level, socialisation, overall health condition, the number of comorbidities), six factors characterising families (respondent’s education level, employment, family income level, the perceived stress level of the child’s primary caretaker, family and informal support) and six factors characterising rehabilitation services (five MOPC subscales and professional support) were used as independent variables.

The final model with nine independent variables explained 61% (adjusted R^2^ = 0.61) of the variance of overall family needs. The final model and the unique influence of each independent variable that had a statistically significant impact on the total number of family needs are presented in Table 11.

#### 3.6.2. Model that Explains Family Needs for Support

The number of needs for support marked by the families (total number of needs in subscales “Needs for Support” and “Family Functioning” in FNS) was used as the dependent variable in this model. Four factors characterising children with cerebral palsy (GMFCS level, CFCS level, socialisation, the number of comorbidities), six factors characterizing the families (respondent’s education level, employment, family income level, the perceived stress level of the child’s primary caretaker, family and informal support) and five factors characterising rehabilitation services (four MOPC subscales and professional support) were used as independent variables in the initial model.

The final model explained 44% of the variance (adjusted R^2^ = 0.44). The final model, and the unique contribution of each independent variable that had a statistically significant impact on the total number of family needs for support, are depicted in Table 12.

#### 3.6.3. Model that Explains Family Needs for Community Services

The number of family needs for community services (“Community Services” subscale in FNS) was used as the dependent variable in this model. Five factors characterising children with cerebral palsy (GMFCS level, CFCS level, socialisation, overall health condition, the number of comorbidities), six factors characterizing the families (respondent’s education level, employment, family income level, the perceived stress level of the child’s primary caretaker, family and informal support), and eight factors characterising rehabilitation services (five MOPC subscales, access to rehabilitation services in the place of residence, the regularity of receiving the rehabilitation services and professional support) were used as independent variables in the initial model.

The final model explained 52% (adjusted R^2^ = 0.52) of the variance of family needs for community services. The end-model, and the unique influence of each independent variable that had a statistically significant impact on the total number of family needs for community services, are depicted in Table 13.

#### 3.6.4. Model that Explains Financial Needs of Families

The number of financial needs for families (“Financial Needs” subscale in FNS) was used as the dependent variable in this model. Four factors characterising children with cerebral palsy (GMFCS level, CFCS level, socialisation, the number of comorbidities), six factors characterising the families (respondent’s education level, employment, family income level, the perceived stress level of the child’s primary caretaker, family and informal support) and eight factors characterising rehabilitation services (five MOPC sections, access to rehabilitation services in the place of residence, the regularity of receiving the rehabilitation services and professional support) were used as independent variables in the initial model.

The final model explained 53% (adjusted R^2^ = 0.53) of the variance of the financial needs of families. The end-model, and the unique contribution of each independent variable that had a statistically significant impact on the total number of financial needs for families, are presented in Table 14.

#### 3.6.5. Model that Explains Family Needs for Information

The number of family needs for information (“Needs for Information” subscale in FNS) was used as the dependent variable in this model. Five factors characterising children with cerebral palsy (GMFCS level, CFCS level, socialisation, overall health condition, the number of comorbidities), six factors characterizing the families (respondent’s education level, employment, family income level, the perceived stress level of the child’s primary caretaker, family and informal support) and five factors characterising rehabilitation services (four MOPC sections, and professional support) were used as independent variables in the initial model.

The final model explained 23% (R^2^ = 0.23) of variance of family needs for information. The final model, and the unique influence of each independent variable that had a statistically significant impact on the total number of family needs for information, are depicted in Table 15.

## 4. Discussion

Our study evaluated the specific needs of families who have preschool children diagnosed with cerebral palsy, and identified several factors influencing these needs. Although our study focused on families in Latvia, the results may be helpful to families, care-givers and public health officials in other locations.

As Latvia does not have a unified register of patients with cerebral palsy, the information about the number of such families living in Latvia was based on information provided by the State Medical Commission for the Assessment of Health Condition and Working Ability, revealing that there were 264 children registered in Latvia who had been given the status of disabled based on the ICD G80 code (cerebral palsy), and who were between two and seven years of age. The research engaged 234 families, which is 88.6% of all potential families. Among the families included in the research, 84 families, or 36%, lived in Riga, 89 families (38%) lived in other Latvian cities, and 61 families (26%) lived in a rural territory. Such a distribution of places of residence allows us to assume that comprehensive information has been obtained about the needs and factors affecting families living in Latvia and raising preschool children with cerebral palsy.

### 4.1. Results of Family Needs Survey

According to the results of the Family Needs Survey, the biggest share of needs was pointed out in the “Needs for Information” subscale. More than half of respondents gave affirmative answers to all statements in this section. There is a similar trend also in other studies, where the needs of families with children with functional disabilities are explored [17,18,19,20,21,22]. This might mean that the majority of families who are raising children with developmental disorders feel that they lack information, and service providers should think how to improve the provision of information to these families. Still, Palisano and colleagues, observing a similar trend in their study, made the assumption that the large number of “Needs for Information” is possibly related to the opinion of the surveyed parents that “there is never too much information”, rather than a true lack of information [20]. Despite this assumption, health care providers should make sure that families receive professional answers to their questions. It is especially important now, when the internet is broadly used to obtain information but often provides confusing, unsubstantiated information about the child’s treatment and rehabilitation opportunities [23].

In our study, a large number of respondents pointed out that they need information about services that are available for their child with cerebral palsy (88.9%) and services that they would need in the future (85%). The data published by other authors are slightly different. Just 54% of parents surveyed by Farmer, 63% of parents surveyed by Ellis and 59% of parents polled by Palisano mentioned that they need information about the presently available services, while 74%, 78% and 68% of parents, respectively, were interested in future services [17,20,21]. Such differences show that families living in Latvia are less informed about services available for their children with developmental disorders, and service providers should make sure that the family is informed about issues important to it.

Families surveyed by us (more often than families surveyed by other researchers) said that they wish to obtain information about their child’s disorders, as well as about training and education opportunities for the child. Possibly, such increased interest can be explained via the peculiarities of the age of the children. Our research included preschool children, while other authors studied families with children of different ages, including school age. It is noted that the younger the child, the higher the parents’ interest in all kinds of information. Young parents are scared and unconfident, they have not yet gotten used to the new situation and supply of services, and they are looking for every opportunity to promote the child’s development [20].

The next most important area, in which most of the families expressed a wish for additional support, was needs related to community services (treatment, rehabilitation, preschools, etc.) and financial support. Such a result is no surprise. It is well-known that cerebral palsy is an “expensive” disorder, and its costs may reach EUR 900,000 throughout a lifetime [24]. The availability of services and financial challenges for families who are raising children with cerebral palsy are much higher than for other families with children of similar age [25]. As is often a case, and was so in our research group, if one parent is not working any more or is working part-time, financial challenges are even higher.

It is alarming that 73% of respondents said that the family needs help to coordinate medical, social and education services. It should be noted that in countries with historically stable social support system, families less often point out the necessity related to the coordinated provision of medical, social and educational services [7,19,20,21].

It is an established fact that as the child with functioning disorders grows, the need for financial support also grows [7]. Since in the families we surveyed the children were up to the age of seven, a comparatively small number of families said that they would need financial assistance for house modification. Still, more than half of the respondents needed financial support for special equipment or assistive devices. A similar need for financial support for the purchase of assistive devices was voiced by parents surveyed by Nitta in Japan and parents surveyed by Wang in China [18,26]. Meanwhile, in Farmer’s and Palisano’s surveys in the US, such needs were marked by just 19% and 34% of parents, respectively [20,21].

A large part of the surveyed families pointed out that they need not only financial, but also moral and psychological, support. As taking care of the sick child takes a large part of the day, and half of the children did not attend a preschool, it is not a surprise that the majority of the surveyed parents would like to have more time to themselves. Such type of needs is marked as important also in the reports published by other authors [17,18,26].

Like in the research by Palisano and colleagues, just a small number of respondents noted needs that are related to family functioning [20]. It is possible that families indeed do not need such assistance. Still, it cannot be excluded that parents are not aware of, or are not admitting, such needs. It is believed that parents who have a child with developmental orders or a chronic illness more often think about how to promote the child’s development and focus less on family needs, or do not consider them as needs, which might be a reason for elevated stress and families breaking apart [27].

### 4.2. Analysis of Factors Explaining Family Needs

As hypothesised, the needs of families raising preschool children with cerebral palsy living in Latvia are affected by factors characterising families, children with cerebral palsy and rehabilitation service providers, and the unique impact of these factors depends on the type of family needs.

The literature sources name the child’s functioning limitations and health condition as significant factors influencing family needs—the more distinct the functioning limitations and complicated the health disorders, the higher are the family needs [8,18,20,21,28,29]. In our research, the level of the child’s mobility limitations was a significant factor influencing the financial needs of families—families whose children were able to walk without any assistive devices marked lower needs for financial support compared to families whose children did not have such a possibility. The obtained data match with information published by other authors [8,20]. Meanwhile, in contrast to results found by Almasri and colleagues, the influence of the child’s mobility limitation levels on family needs for services and support was not discovered in our research [8].

An unexpected result was the influence of the child’s communication limitations on family needs for community services—families with children who had more distinct communication problems most often noted that they need assistance in the finding and provision of medical, rehabilitation or education services. Obviously, rehabilitation and education services for children with motor disorders in Latvia are more available and better developed than services for children who, in addition, have distinctive communication limitations that often are combined with cognitive disorders; thus, the need for specific education and rehabilitation programmes increases.

The child’s communication limitations also affected the families’ needs for information—more distinct communication disorders in the child increased the number of family needs in this type of needs. It has been established that cognitive and behavioural disorders in children are factors that increase the family needs [8,19], while children with distinctive communication problems often have cognitive limitations and limited socialisation [30,31]. Considering the child’s age, the cognitive level of the children included in the research was not evaluated. Still, an assumption could be made that children with distinctive communication problems had more distinct cognitive limitations that possibly determined the higher family needs for services and information.

The data analysis in our research discovered that as the number of comorbidities associated with brain damage grows, the total family needs increase, as do needs for support and financial needs, and this factor had a significant effect on family needs in the above-mentioned types of needs.

The support provided by the closest family members was a very significant factor, affecting family needs—as the child’s primary caretaker received support and assistance from other family members, the number of family needs declined. Further, other researchers confirmed that families that have good and supportive mutual relations are more successful in solving issues that are related with the care and treatment of the sick child, and these families less often need “external” support [20,32]. This is valuable information for service providers. Service providers should be aware that in cases when the service is received by a family that lacks this internal support, there is possibly a greater need for services, information, and social and financial support.

Such factors as the education level and perceived stress level of the child’s primary caretaker had a similar impact on family needs. A high perceived stress level and elementary education were significant factors increasing family needs for support and community services. Lower education levels of parents so far have not been related with needs for additional support [8,19], while a correlation between elevated stress levels and the increased necessity for support has also been revealed in other studies [21,32,33,34]. It is interesting that the perceived stress level of the child’s primary caretaker had a higher unique impact on the family needs for services than the caretaker’s employment or the family’s income level. Employment and the family’s financial situation often are identified as factors that influence the family’s needs for community services [19,20,21], while we did not find information about the impact of the caretaker’s stress level on this type of needs. Possibly, it is harder for persons with higher stress levels to organise their everyday activities and set priorities, and this has a negative effect on the quality of the person’s life, creating challenges for the optimum planning and organisation of the tasks [32,33].

If there is a child with health and functioning disorders in a family, then the availability of different health and social services becomes important. Farmer and Almasri in their studies discovered that families who live in cities where health, social care and educational institutions are more easily accessible in general mark lower needs than families living in more distant regions [8,21]. Our research also revealed a similar trend—families living in Riga noted lower needs for community services than families living in other cities or rural territories. Still, the influence of this factor on this area of needs was not confirmed. In our research, neither the place of residence nor the number of children in the family had a significant impact on family needs.

Among the most significant factors reducing family needs are higher socio-economic status and higher income [7,8,21,35]. This information is also confirmed in our research. Medium and, even more so, low family income levels increased the total number of family needs, and the number of family needs for community services, information and financial support. The only type of needs reviewed in the study that was not affected by the income level was family needs for support—equal needs for formal and informal support were voiced by those respondents whose family income level was assessed as high and those who assessed their income level as medium or even low.

Surprisingly, but in contrast with information published by Almasri and colleagues [8], neither the availability of rehabilitation services at the place of residence nor the regularity of receiving rehabilitation services were factors that affected the family needs of our respondents. Support received from professionals, however, turned out to be a significant factor reducing needs. Higher support from professionals was a significant factor reducing total family needs and needs for information and support. Further, there was a positive reducing effect on family needs if the family noted that the received services complied with the principles of family-centred care. In particular, the role of the principle “Enabling and Partnership” should be underscored. By providing services based on cooperation and partnership principles, i.e., engaging the family in decision making as an equal partner, it is possible to significantly reduce the overall family needs and needs for support, services and additional financing. This is significant information for service providers, which proves the importance of the way the family is engaged in the treatment and rehabilitation processes, and of the skills of service providers in communicating with the client and cooperating. Our results match with the reports by Palisano and Almasri on the positive impact of family-centred care on reducing family needs in families who are raising children with cerebral palsy [8,20].

### 4.3. Summary of Factors Explaining Family Needs

In our research, 6 factors characterising children with cerebral palsy, 10 factors characterising families and 9 factors characterising rehabilitation services were reviewed as possible factors influencing family needs, but only one of the analysed factors—the family’s internal support—was identified as a significant factor influencing family needs in all areas of needs. As the child’s primary caretaker received support from other family members, other needs were considerably reduced.

Like in the reports of other authors [8,19,20], our research also revealed that factors reducing family needs included such socio-economic factors as the family’s income level, which was a significant factor affecting family needs in four types of needs, and the employment of the child’s primary caretaker, which had an impact on three types of family needs. The impact of factors related to the child’s health condition was less important—a larger number of comorbidities was a factor increasing family needs in three areas of needs, but its unique impact was low. Mobility and communication limitations were factors increasing family needs only in two areas of needs—distinctive communication limitation for the child increased family needs for community services and information, while severe mobility limitations increased financial needs and overall family needs.

The perceived stress level of the child’s primary caretaker turned out to be a significant factor affecting family needs in three areas of needs—higher perceived stress level increased the overall family needs and needs for community services and support.

The above-mentioned factors are unique, individual and in most cases cannot be directly influenced. Service providers who work with families raising a preschool child with cerebral palsy should consider the impact of the above-mentioned factors on family needs, and should pay additional attention to families with potentially higher risks of needs.

Analysis of the research results also revealed the significant impact of those factors that directly depend on service providers. Thus, the provision of rehabilitation services based on the family-centred care principle “Enabling and Partnership” was a significant factor reducing family needs in four areas of needs—overall family needs, needs for support, services and financial support. Meanwhile, a family receiving more support from professionals who work with the child or the family was a significant factor reducing family needs in three areas of needs.

In conclusion, the most important factors affecting the needs of families living in Latvia are related with the socio-economic situation and support of peers—family members or professionals who work with the family. Limitations in the child’s functioning and health disorders were factors influencing family needs to a lower extent.

### 4.4. Methodological Analysis and Limitations of the Research

In order to achieve the objective of the research, an analytical cross-sectional design was selected for the study. Thus, information about families living in Latvia and raising preschool children diagnosed with cerebral palsy, as well as family needs and factors affecting them, was obtained during a certain period of time. In order to claim that the information obtained during the study can be generalised in relation to another time period, data should be obtained repeatedly, but this was not planned. Thus, information obtained about the needs of families living in Latvia and factors affecting them should be assessed and interpreted with caution, not excluding possible changes in time.

Family needs were identified using a standardised questionnaire, the Family Needs Survey, which, during the period of obtaining the data, was the most recognised and broadly used questionnaire for identifying family needs. The questionnaire was developed to learn about the needs of those families who are raising preschool children with development disorders. Still, the questionnaire was developed several dozens of years ago and, even though its modified version was used to obtain the data, it is still possible that not all the needs of families living in Latvia and raising children with cerebral palsy were identified. The advantage of using a standardised questionnaire is simplicity in obtaining data, and the opportunity to analyse these data by using quantitative data analysis methods. Still, such a way of obtaining data does not allow one to study the problem thoroughly; therefore, it is recommended to conduct a more thorough study of the obtained data, using qualitative methods for data extraction and analysis.

The possible factors influencing family needs were identified, based on a comprehensive study of literature sources and factors that may affect family needs in cases when families take care of children with cerebral palsy. The needs of families raising children with developmental disorders have been studied broadly, but, during the preparation phase, we did not manage to find sources that have studied family needs in the nearest geopolitical region, with similar historical and socio-economic backgrounds. Thus, it is possible that we did not manage to identify the type of needs or possible factors influencing these needs that are unique for our region. As a limitation to the study, we should also name the fact that the overall number of comorbidities of the child was viewed as a factor affecting family needs, without assessing the possible impact of each comorbidity separately.

This is proven in the multiple linear regression analysis—the final models explained 23–61% of the changes in family needs. Thus, there could be other unpredicted and unidentified factors that might affect the needs of the families living in Latvia. Another limitation of the study is the fact that information characterising the child’s health condition was obtained from the caretaker, and we did not assess the possible unique impact of particular elements characterising the health conditions, such as behavioural problems, on family needs.

A more targeted approach would be to use the qualitative research design, which would allow us to understand the research problem more thoroughly and present a new hypothesis.

The factors affecting family needs were determined using the multiple linear regression analysis method. Such an analysis method was selected because it allows one to study the dependence of the features on a number of independent features, and the data obtained in the study met the requirements for conducting a multiple linear regression analysis. However, this type of analysis has its drawbacks. Even though we managed to study factors that might affect family needs and test their impact on the stability of the regression models, still, some inaccuracies during the data obtaining and analysis process cannot be excluded. It is also established that not all independent variables have a direct impact on the dependent feature [20]; therefore, for a deeper understanding of the research problem and construct, further studies should focus on the indirect impact of the identified (and possible new) factors.

## 5. Conclusions

Descriptive analysis showed that families living in Latvia most often need information, social and financial support and coordination of services, and they also need financial support to cover the costs of child care and treatment. The results of the data analysis support the hypothesis that factors characterising families, children with cerebral palsy and rehabilitation services affect the needs of the families with preschool children with cerebral palsy living in Latvia, and the unique impact of these factors depends on the type of needs. Regression analysis revealed that the most important factors affecting the needs of families were related with the socio-economic situation, as well as the support of peers and professionals. The availability and regularity of rehabilitation services, limitations to the child’s functions and health impairments were factors that affected family needs to a lesser extent.

## Figures and Tables

**Figure 1 jpm-10-00139-f001:**
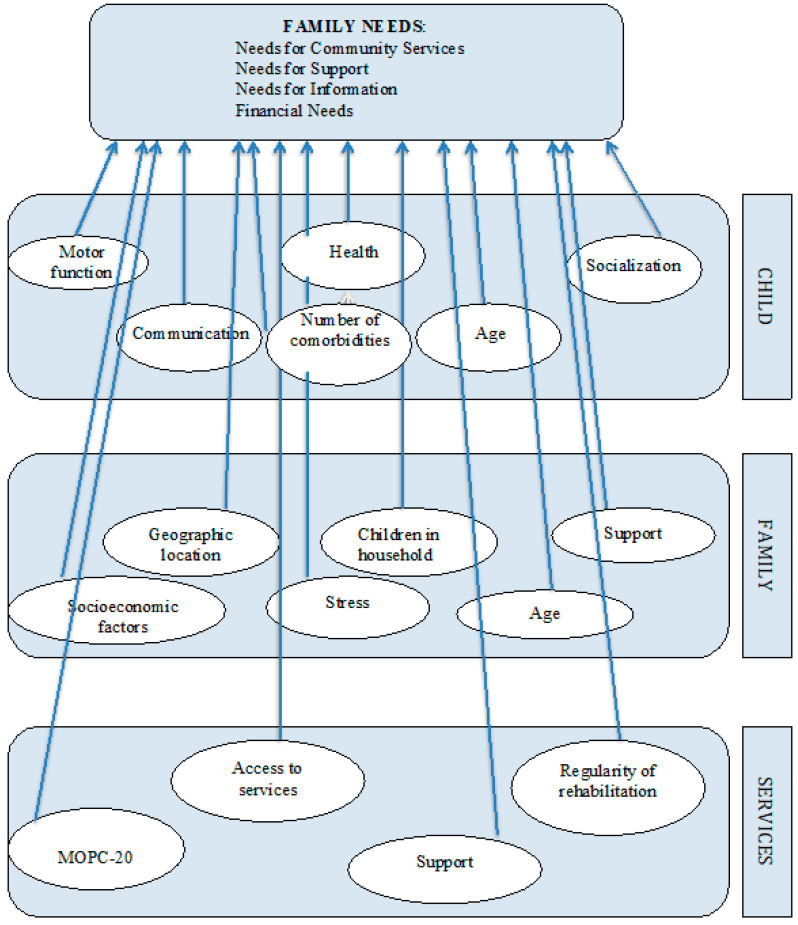
Model of the Determinants of Family Needs.

**Table 1 jpm-10-00139-t001:** The level description of Gross Motor Function Classification System (GMFCS).

Level	Description
I	Walks without limitation
II	Walks with limitation
III	Walks with assistive mobility devices
IV	Self-mobility with limitation, children are transported or use wheeled mobility
V	Self-mobility is severely limited, children are transported

**Table 2 jpm-10-00139-t002:** The level description of Communication Function Classification System (CFCS).

Level	Description
I	Sends and receives information with familiar and unfamiliar partner effectively
II	Sends and receives information with familiar and unfamiliar partner—may need extra time
III	Sends and receives information with familiar partner but not with unfamiliar partners
IV	Inconsistently effectively sends and receives information even with familiar partner
V	Seldom effectively sends and receives information even with familiar partner

**Table 3 jpm-10-00139-t003:** Data used in study analysis and their description.

Variable	Measures Used to Obtain Data, Type of Data
**OUTCOME MEASURES**	
Family Needs (overall)	FNS: items 0–41
Needs for Information	FNS: items 0–9
Needs for Support	FNS: items 0–13
Financial Needs	FNS: items 0–7
Needs for Community Services	FNS: items 0–6
**INDEPENDENT VARIABLES**	
***Child variables***	
Age	Questionnaire: years
Motor function	GMFCS: items 1–5
Communication function	CFCS: items 1–5
Comorbidities (total number)	Questionnaire: number of comorbidities 0–6
Child’s health (parent reported)	Questionnaire: rather good/rather poor
Socialisation	Questionnaire: preschool/no preschool
***Family variables***	
Age	Questionnaire: years
Employment	Questionnaire: employed/not employed
Education	Questionnaire: lower than secondary school/secondary school/bachelor’s or master’s degree
Marital status	Questionnaire: married or living with partner/single
Family income level (self-report)	Questionnaire: high/average/low
Children in household	Questionnaire: one/two or more
Geographic location	Questionnaire: Riga/urban/rural
Stress	PSS-10: items 10–50
Family support	FSS: items 6–30
Informal support	FSS: score 7–35
***Service variables***	
Formal support	FSS: score 5–25
Access to rehabilitation services	Questionnaire: yes/no
Regularity of rehabilitation	Questionnaire: regularly/at least twice a year/once a year
Family-centredness of services	
Enabling and Partnership	MOPC-20: items 3–21
Providing General Information	MOPC-20: items 5–35
Providing Specific Information	MOPC-20: items 3–21
Co-ordinated Care	MOPC-20: items 4–28
Respectful and Supportive Care	MOPC-20: items 5–35

**Table 4 jpm-10-00139-t004:** Characteristics of participants and their families.

Characteristics	n	%	95% CI
**Relationship to the child**			
Mother	218	93.2	89.9–96.4
Father	4	1.7	0.5–3.3
Grandmother	8	3.4	1.1–5.7
Guardian	4	1.7	0.5–3.3
**Education**			
Bachelor’s/master’s degree	92	39.3	33–45.6
Secondary school	122	52.2	45.6–58.6
Lower than secondary school	20	8.5	4.9–12.1
**Employment**			
Employed	117	50	43.6–56.5
Not employed	117	50	43.6–56.5
**Marital status**			
Married or living with partner	192	82.1	77.1–87
Single	42	17.9	13–22.9
**Children in household**			
One	115	49.1	42.7–55.6
Two or more	119	50.9	44.8–56.1
**Geographic location**			
Rīga (capital city)	84	35.9	29.7–42.1
Urban (any other city)	89	38.0	31.7–44.3
Rural	61	26.1	20.4–31.7
**Family income (EUR per month)**			
Less than 420	42	18.0	13.9–23.9
420–839	136	58.1	49.2–63.1
840–1120	39	16.6	10.1–21.9
More than 1120	17	7.3	3.9–12.7
**Family income level ***			
Low	44	18.8	14.1–24.2
Average	157	67.1	61.1–73
High	33	14.1	9.6–18.6

* Parent—reported.

**Table 5 jpm-10-00139-t005:** Characteristics of the children with cerebral palsy.

Characteristics	n	%	95% CI
**Gender**			
Male	131	55.6	49.1–61.9
Female	105	44.4	38–50.8
**Comorbidities ***			
Visual impairment	93	39.7	33.4–45.9
Hearing impairment	28	12.0	7.8–16.1
Cognitive impairment	158	67.5	61.5–73.5
Behaviour disturbance	24	9.7	6.8–12.2
Seizure	58	24.8	19.3–29.9
**Child health ***			
Rather good	94	39.7	36.2–43.1
Rather poor	142	60.3	56.1–64.8
Socialisation			
Yes (pre–school etc.)	140	59.4	53.1–65.2
No (home)	96	40.6	36.2–44.3
**Type of cerebral palsy**			
Spastic unilateral	77	32.9	26.8–28.9
Spastic bilateral	112	47	40.5–53.4
Dyskinetic	25	10.7	6.7–14.6
Ataxic	9	3.8	1.4–6.3
Not specified/mixed	13	5.6	2.6–8.5
**GMFCS level**			
I	78	33.3	27.3–39.1
II	45	19.2	14.7–23.8
III	44	18.1	14.4–21.7
IV	49	20.9	16.7–25.1
V	20	8.5	5.1–11.9
**CFCS level**			
I	55	23.5	19.3–27.8
II	42	17.9	13.5–21.9
III	45	18.4	14.3–22.3
IV	56	23.9	19.4–28.4
V	38	16.2	12.4–20

* Parent—reported.

**Table 6 jpm-10-00139-t006:** The results of the Family Needs Survey.

Need	n *	%
**Needs for Information**		
I need more information about how children grow and develop	125	53.4
I need more information about my child’s condition or disability	157	67.1
I need more information about how to play with or talk to my child	125	53.4
I need more information about how to teach my child	178	76.1
I need more information about how to handle my child’s behaviour	143	61.1
I need more information on the services that are presently available for my child	208	88.9
I need more information about the services that my child might receive in the future	199	85.0
I need more information about planning my child’s future wellbeing	167	71.4
I need help in finding information about future education for my child	200	85.5
**Needs for Support**		
I need to have someone in my family that I can talk to more about problems	45	19.2
I need to have more friends that I can talk to	63	26.9
I need to have more opportunities to meet and talk with parents of children who have disabilities	137	58.5
I need reading material about other parents who have a child similar to mine	169	72.2
I need to have more time for myself	149	63.7
I need to have more time just to talk with my child’s teacher or therapist	105	44.9
I need to talk more to a minister who could help me deal with problems	51	21.8
I would like to meet more regularly with a counsellor (psychologist, social worker, psychiatrist) to talk about problems	105	44.9
**Explaining to Others**		
I need more help in explaining my child’s condition to either my parents or my spouse’s parents	34	14.5
I need more help in explaining my child’s condition to my spouse	21	9.0
I need more help in how to explain my child’s condition to his/her siblings	19	8.1
I need help in explaining my child’s condition to other children	78	33.3
I need help in knowing how to respond when friends, neighbours, or strangers ask questions about my child’s condition	79	33.8
I need help in explaining my child’s condition to teachers and other professionals	68	29.1
**Community Services**		
I need help in locating a child care centre or preschool for my child	108	46.5
I need help locating babysitters or respite care providers who are willing and able to care for my child	74	31.6
I need help in getting appropriate care for my child in our church or synagogue during services	3	1.3
I need help locating a doctor who understands me and my child’s needs	80	34.2
I need help in locating rehabilitation services for my child	143	61.1
I need help in coordinating medical, developmental, educational, and other community services for my child	170	72.6
**Financial Needs**		
I need more help in paying for expenses such as food, housing, medical care, clothing, or transportation	92	39.3
I need more help in paying for special equipment that my child needs	134	57.3
I need more help in paying for therapy, child care, or other services that my child needs	180	76.9
I need more help in paying for babysitting or respite care	82	35.0
I need more help in paying for home modifications	68	29.1
I need more help in paying for toys that my child needs	24	10.3
I or my spouse need more counselling or help in getting a job	45	19.2
**Family Functioning**		
My spouse needs help in understanding and accepting our child’s condition	12	5.1
Our family needs help in discussing problems and reaching solutions	40	17.1
Our family needs help in learning how to support each other during difficult times	61	26.1
Our family needs help in deciding who will do household chores, child care, and other family tasks	3	1.3
Our family needs help in deciding and doing recreational activities	39	16.7

* Number of respondents with mark “3” (I definitely need help with this).

**Table 7 jpm-10-00139-t007:** The results for subscales of the Family Needs Survey.

Subscale	Items in Subscale	Min	Max	Mean	SD
Needs for Information	9	0	9	6.5	2.0
Needs for Support	8	0	8	3.5	2.1
Explaining to Others	6	0	6	1.3	1.5
Community Services	6	0	6	2.5	1.3
Financial Needs	7	0	7	2.7	1.4
Family Functioning	5	0	5	0.7	0.9

**Table 8 jpm-10-00139-t008:** Statistically significant correlations between family needs and child’s characteristics.

	Needs for Information	Needs for Support	Community Services	Financial Needs	Family Needs (Total)
Age	-	-	-	-	-
GMFCS level	0.10 *	0.17 *	0.45	0.44	0.34
CFCS level	0.34	0.24	0.46	0.37	0.42
Socialisation	0.16 *	0.22	0.42	0.26	0.31
Child health	0.15 *	-	-	-	0.19 *
Number of comorbidities	0.28	0.36	0.37	0.36	0.46

*p* < 0.001; * *p* < 0.05.

**Table 9 jpm-10-00139-t009:** Statistically significant correlations between family needs and family characteristics.

	Needs for Information	Needs for Support	Community Services	Financial Needs	Family Needs (Total)
Age	-	-	-	-	-
Marital status	-	-	-	-	-
Education	0.16 *	0.19 *	-	-	0.20 *
Employment	0.21	0.19 *	0.33	0.35	0.34
Family income level	0.31	0.29	0.36	0.46	0.43
Geographic location	-	-	0.19 *	-	-
Children in household	-	-	-	0.15 *	-
Stress	0.22	0.30	0.15 *	0.22 *	0.33
Support from family	−0.23	−0.44	−0.18 *	−0.27	−0.39
Informal support	−0.28	−0.26	−0.13 *	−0.19 *	−0.31

*p* < 0.001; * *p* < 0.05.

**Table 10 jpm-10-00139-t010:** Statistically significant correlations between family needs and service characteristics.

	Needs for Information	Needs for Support	Community Services	Financial Needs	Family Needs (Total)
Respectful and Supportive Care	−0.19 *	−0.23	−0.30	−0.27	−0.30
Enabling and Partnership	−0.28	−0.35	−0.32	−0.36	−0.40
Co-ordinated and Comprehensive Care	−0.22	−0.25	−0.29	−0.25	−0.31
Providing General Information	−0.20 *	−0.17 *	−0.18 *	−0.21	−0.25
Providing Specific Information	−0.14 *	-	−0.21 *	−0.14 *	−0.16 *
Formal support	−0.34	−0.35	−0.29	−0.29	−0.43
Regularity of rehabilitation	-	-	0.19 *	0.13 *	-
Access to rehabilitation services	-	-	0.14 *	0.14 *	-

*p* < 0.001; * *p* < 0.05.

**Table 11 jpm-10-00139-t011:** Multiple regression model explaining overall family needs.

Regression Model: F (9, 233) = 39.18; *p* = 0.000						
Variable	B	SE	β	t	*p*	% of Unique Contribution
Income level						2.3
Low vs. high	4.23	1.14	0.24	3.882	0.000	
Average vs. high	1.96	0.88	0.13	2.355	0.032	
Employed vs. not employed	2.52	0.63	0.19	4.151	0.000	2.8
Support from family	−0.58	0.08	−0.39	−6.835	0.000	8.2
Stress	0.18	0.04	0.18	4.245	0.000	3.0
GMFCS Level	0.51	0.22	0.11	2.261	0.040	0.7
Number of comorbidities	0.85	0.22	0.19	3.932	0.000	2.5
Enabling and Partnership	−1.18	0.25	−0.22	−4.872	0.000	3.9
Formal support	−0.56	0.12	−0.21	−4.556	0.000	3.5

**Table 12 jpm-10-00139-t012:** Multiple regression model, explaining family needs for support.

Regression Model: F (7, 233) = 26.65; *p* = 0.000						
Variable	B	SE	β	t	*p*	% of Unique Contribution
Education						2.1
Lower than secondary school vs. secondary school	1.29	0.49	0.13	2.568	0.011	
Bachelor’s or master’s degree vs. secondary school	0.21	0.14	0.04	0.715	0.475	
Stress	0.06	0.02	0.16	3.164	0.002	2.7
Enabling and Partnership	−0.46	0.11	−0.22	−4.126	0.000	4.5
Support from Family	−0.28	0.03	−0.37	−7.422	0.000	13.9
Formal support	−0.23	0.05	−0.21	−4.011	0.000	4.2
Number of comorbidities	0.34	0.09	0.19	3.603	0.000	3.5

**Table 13 jpm-10-00139-t013:** Multiple regression model explaining family needs for community services.

Regression Model: F (8, 233) = 32.23; *p* = 0.000						
Variable	B	SE	β	t	*p*	% of Unique Contribution
CFCS level	0.32	0.06	0.28	5.341	0.000	5.8
Preschool vs. no preschool	0.72	0.17	0.25	4.244	0.000	3.7
Income level						2.2
Low vs. high	0.98	0.28	0.23	3.375	0.002	
Average vs. high	0.45	0.21	0.13	1.958	0.051	
Employed vs. not employed	0.48	0.16	0.17	3.112	0.007	1.7
Stress	0.05	0.01	0.18	3.920	0.001	2.8
Support from Family	−0.61	0.02	−0.12	−2.466	0.024	1.4
Enabling and Partnership	−0.29	0.06	−0.22	−4.102	0.000	3.5

**Table 14 jpm-10-00139-t014:** Multiple regression model explaining financial needs for families.

Regression Model: F (7, 233) = 34.29; *p* = 0.000						
Variable	B	SE	β	t	*p*	% of Unique Contribution
GMFCS level	0.34	0.06	0.28	5.244	0.000	5.8
Number of comorbidities	0.14	0.05	0.12	2.206	0.046	0.9
Employed vs. not employed	0.63	0.16	0.20	3.936	0.000	3.1
Income level						10.8
Low vs. high	1.96	0.29	0.45	6.498	0.000	
Average vs. high	0.77	0.23	0.17	2.595	0.017	
Enabling and Partnership	−0.273	0.06	−0.21	−4.012	0.000	3.5
Support from Family	−0.09	0.02	−0.17	−3.294	0.002	2.4

**Table 15 jpm-10-00139-t015:** Multiple regression model, explaining family needs for information.

Regression Model: F(6, 233) = 12.23; *p* = 0.000						
Variable	B	SE	β	t	*p*	% of Unique Contribution
CFCS level	0.02	0.01	0.17	2.865	0.005	2.5
Income level						3.5
Low vs. high	0.12	0.03	0.28	3.280	0.001	
Average vs. high	0.10	0.03	0.26	3.122	0.002	
Providing General Information	−0.03	0.01	−0.16	−2.637	0.009	2.1
Formal support	−0.02	0.01	−0.20	−3.218	0.001	3.2
Support from Family	−0.01	0.02	−0.17	−2.618	0.009	2.0

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
