# Peer review of "Needs of Families with Children with Cerebral Palsy in Latvia and Factors Affecting These Needs"

_jpm, 2020, doi:10.3390/jpm10030139_

Round 1

Reviewer 1 Report

The paper entitled “Needs of families with children with cerebral palsy in Latvia and factors affecting these needs” report on an interesting issue on the needs of the families with preschool children with cerebral palsy. The study is well written and give very useful information for clinicians involved in the care of family and children with CP

I have few concerns

Introduction

Line 22: Please add a better definition of cerebral palsy (see Bax et al 2005)

Methods

Line 51: delete “of”

You included children between 2-7 years described as pre-school age children. Are children of 6-7 years considered pre-scooler in your country? This should be reported in the text

More information on clinical characteristics of the children should be reported; for example the presence of cognitive impairment or epilepsy or other behavioral problems that could affect the needs of families

Results

Did you find any difference between the fathers and the mothers? Who completed the questionnaires?

Table 7 and 8: it is not clear if the data are statistically significant or not. Are you sure that a correlation of 0.10 had a p<0.05?

Discussion

The authors reported that child’s health condition (functional limitation) had a low individual impact on family needs. This data is not correctly interpretable without further information on the study population; most authors described that in children with CP the presence of behavioral problems influence the quality of life of the family more than motor functional impairments (see Brehaut 2004, Eker 2004, Raina 2005, Romeo et al 2010) and therefore could influence the needs of the family.

Reviewer 2 Report

Thank you for the opportunity to review this manuscript. This is an interesting study aimed to identify the needs of the families with preschool children with cerebral palsy. While this work is adequately motivated I feel that the underlying study design needs significant amounts of work before it can be published. There are the main shortcomings of the study that need major revision.

My concerns are outlined below:

1) The abstract briefly defines the aim of the study and main conclusions, however, in the results section only a single sentence (which is basically a conclusion) was placed, without the fundamental results. Please revise.

2) Overall, the paper is difficult to follow due to the inaccuracy of language, multiple typos which are easily identifiable using spell-check and inadequate grammar. This manuscript needs to be edited for English and grammar.  The authors need to correct the manuscript to reflect person-first language.

3) Please provide more detail how the sample size was determined. Was an a priori power analysis performed?  If not, I suggest the authors to add a power analysis to verify the sample size.

4) The second part of the aim of this paper: 'and to study how these needs relate with the factors characterizing families, children and rehabilitation services' need to be more clearly defined. It is reads like the study was conducted and the data analyzed without clear hypotheses. If the study was exploratory, then that needs to be clearly stated. This is a major weakness. It was not clear what the real underlying purpose. The aim of this paper needs to be more clearly defined.

5) There are several nonparametric measures of relationships based on the similarity of ranks in two variables (e.g., multiple regression, principal component analysis, multiple correspondence analysis). Nevertheless, canonical correlation analysis CCA as a new technique in the clinical domain seems the most appropriate procedure to investigate the relationship between two sets of variables for your study. Have you considered using CCA? It would beneficial to consult a statistician on the non-parametric analysis.

6) Overall, the discussion section reads more like the results section. They summarize the tables but do not discuss or compare and contrast findings with previous literature.  This section needs more critical thinking, further analyses and depth. Authors should highlight more the originality of the contribution of their study to literature.

Round 2

Reviewer 2 Report

The authors have made enough revisions to their manuscript.

Author Response

Dear sir/madam

Thank you very much for the review of our manuscript entitled: “Needs of families with children with cerebral palsy in Latvia and factors affecting these needs”.  We sincerely appreciate all valuable comments and suggestions, which helped us to improve the quality of the article.